# Management of Patients Affected by Giant Cell Arteritis during the COVID-19 Pandemic: Telemedicine Protocol TELEMACOV

**DOI:** 10.3390/jpm13040620

**Published:** 2023-03-31

**Authors:** Simone Parisi, Maria Chiara Ditto, Rossella Talotta, Angela Laganà, Clara Lisa Peroni, Enrico Fusaro

**Affiliations:** 1Rheumatology Unit, Azienda Ospedaliero Universitaria Città della Salute e della Scienza di Torino, Corso Bramante 88/90, 10126 Turin, Italy; 2Rheumatology Unit, Department of Clinical and Experimental Medicine, University Hospital Gaetano Martino, 98124 Messina, Italy

**Keywords:** giant cell arteritis, telemedicine, COVID-19, TELEMACOV

## Abstract

Giant cell arteritis (GCA) is the most common primary systemic vasculitis in western countries, prevalently affecting elderly people. Both early diagnosis and regular monitoring are necessary for the correct management of GCA. Following the outbreak of the COVID-19 pandemic, government decisions aiming at reducing the contagion led to reductions in health activities, limiting them to urgent cases. At the same time, remote monitoring activities have been implemented through telephone contacts or video calls carried out by specialists. In line with these deep changes affecting the worldwide healthcare system and in consideration of the high risk of GCA morbidity, we activated the TELEMACOV protocol (TELEmedicine and Management of the patient affected by GCA during the COVID-19 pandemic) in order to remotely monitor patients affected by GCA. The aim of this study was to evaluate the effectiveness of telemedicine in the follow-up of patients already diagnosed with GCA. This was a monocenter observational study. Patients with a previous diagnosis of GCA admitted to the Rheumatology Unit of the University Hospital “Città della Salute e della Scienza” in Turin were monitored every 6–7 weeks by means of video/phone calls from 9 March to 9 June 2020. All patients were asked questions concerning the onset of new symptoms or their recurrence, exams carried out, changes in current therapy, and satisfaction with video/phone calls. We performed 74 remote monitoring visits in 37 GCA patients. Patients were mostly women (77.8%) and had a mean age of 71.85 ± 9.25 years old. The mean disease duration was 5.3 ± 2.3 months. A total of 19 patients received oral glucocorticoids (GC) alone at the time of diagnosis with a daily dose of 0.8–1 mg/kg (52.7 ± 18.3 mg) of prednisone, while 18 patients were treated with a combination of oral steroids (at the time of diagnosis, the prednisone mean dose was 51.7 ± 18.8 mg) and subcutaneous injections of tocilizumab (TCZ). During the follow-up, patients additionally treated with TCZ reduced their GC dose more than patients treated with GC alone (*p* = 0.03). Only one patient, who was treated with GC alone, had a cranial flare and needed to increase the dosage of GC, which led to rapid improvement. Furthermore, all patients proved very adherent to the therapies (assessed by Medication Adherence Rating Scale (MARS)) and considered this type of monitoring very satisfactory according to a Likert scale (mean score 4.4 ± 0.2 on a 1–5 range). Our study shows that telemedicine can be safely and effectively used in patients with GCA under control as a possible alternative, at least for a limited period of time, to traditional visits.

## 1. Introduction

Giant cell arteritis (GCA) is the most common primary systemic vasculitis in western countries, with a lifetime risk of 1.0% for women and 0.5% for men aged more than 50 years [1]. The highest incidence is among people aged 70–79 years [2].

GCA is a chronic disease characterized by a higher morbidity rate. The main symptoms consist of headache, scalp tenderness, and temporal artery abnormalities, such as thickening, tenderness and/or pulselessness. Systemic manifestations such as polymyalgia symptoms, weight loss, fatigue, and fever may be present. In the late stage of the disease, aneurysm and desiccation of the thoracic and abdominal aorta may develop [3].

GCA diagnosis is primarily made on the basis of a positive anamnesis and clinical examination. A recent-onset headache, abnormalities in the temporal artery detected by visual inspection or palpation, and systemic symptoms are highly indicative of the disease. Moreover, more than 95% of GCA cases at diagnosis have an increase in serum markers of inflammation, such as the erythrocyte sedimentation rate (ESR) and C-reactive protein (CRP). The monitoring of these laboratory parameters is particularly useful in the follow up of GCA patients and in the evaluation of the response to therapies. The diagnosis of cranial GCA can be further confirmed by temporal artery biopsy revealing the typical histopathological features of temporal arteritis. Typical demographic, clinical, laboratory, and histopathological features have been altogether included in a core set of classification criteria elaborated in 1990 by the American College of Rheumatology (ACR) [4].

Though not included in these criteria, imaging, such as ultrasound examination, computerized tomography (CT) scan, magnetic resonance imaging (MRI), or positron emission tomography (PET), can also be helpful in the clinically suspected diagnosis of GCA [5,6,7].

For many years, the treatment of GCA has relied on the sole use of glucocorticoids (GC). Recently, an anti-interleukin 6 monoclonal antibody, tocilizumab (TCZ), has been introduced in the therapeutic algorithm of GCA based on its steroid-sparing effect and efficacy in reducing flare rates [6,7,8].

Given the cumulative toxicity of long-term treatment with steroids administered at medium to high doses, GCA therapy needs to be constantly remodulated according to disease activity and/or potential side effects. According to the European League Against Rheumatism (EULAR) guidelines, patients with low disease activity should gradually reduce their GC dose until the final discontinuation [9]. Therefore, in order to achieve good disease activity control and thus have a better prognosis, GCA patients must be diagnosed early and constantly monitored.

In 2018, the EULAR Committee elaborated a minimal dataset of clinical and instrumental data to be kept in consideration for the periodic follow-up of GCA patients in clinical studies and real-life practice [10]. Some of these data can be recorded by means of a phone interview, without requiring a clinical examination.

The pandemic, caused by the Severe Acute Respiratory Syndrome CoronaVirus-2 (SARS-CoV-2) outbreak in China in 2019 and rapidly spreading across continents in 2020, has forced international governments to apply several restrictions in order to limit the spread of the disease. Social distancing and other protective measures have had enormous repercussions on the healthcare system, limiting daily activities to urgencies and reducing or even deleting the periodic follow-up visits of chronically ill patients.

Since the declaration of the global emergency status on 11 March 2020 by the World Health Organization (WHO), ours and many other Italian Rheumatology Units have been forced to discontinue the follow-up visits of rheumatic patients, including those of subjects affected by potentially life-threatening diseases such as GCA.

At the same time, remote monitoring activities have been implemented through telephone contacts or video calls carried out by specialists or based on patient demand. Following this line, a new remote service named Telehealth was developed with the aim of delivering healthcare services under the circumstances in which patients and providers are separated by distance. Telehealth, according to the definition of the WHO, “uses information and communications technology (ICT) for the exchange of information for the diagnosis and treatment of diseases and injuries, research and evaluation, and for the continuing education of health professionals. Telehealth can contribute to achieving universal health coverage by improving access for patients to quality, cost-effective health services wherever they may be. It is particularly valuable for those in remote areas, vulnerable groups and ageing populations” [11].

During the coronavirus disease 2019 (COVID-19) pandemic, Telehealth would have represented a valuable means of breaking down distances between a referral center and patients.

Consequently, our unit activated the TELEmedicine and Management of patients with giant cell Arteritis during the COVID-19 pandemic (TELEMACOV, protocol number: 00167/2020) protocol for rheumatic patients with GCA, in line with the Chronic Care Model [12,13], which provides a relationship model between an informed patient and a medical team involved in healthcare decisions. Patients would be remotely monitored and followed up through telemedicine in order to constantly adjust the current therapies according to reported disease activity and prevent the risk of GCA relapse.

In this study, we aimed to evaluate the efficacy, safety, patient compliance, and satisfaction toward this intervention in a cohort of GCA Italian patients, who were remotely followed up for three months.

## 2. Materials and Methods

### 2.1. Population

Patients with an established diagnosis of GCA from ≤1 year, according to the 1990 ACR criteria [7], and admitted to the Rheumatology Unit of the University Hospital “Azienda Ospedaliera-Universitaria Città della Salute e della Scienza di Torino” were continuously recruited, provided that they did not have any cognitive deficit or hearing loss that could have compromised the reliability of the information obtained by video/phone interviews.

Patients were monitored every 6–7 weeks by means of video/phone calls lasting 20–30 min, from 9 March 2020 to 9 June 2020. Both the patients and specialists gave prior informed consent to the collection of data recorded during the video/phone call.

All patients were asked questions concerning the onset of new symptoms or their recurrence, exams carried out, changes in current therapy, and satisfaction with the video/phone call (Table 1). During the follow up, the patients underwent blood chemistry tests to monitor the progress of the disease and the tolerability of the pharmacological therapy, as indicated by the guidelines [10].

The degree of satisfaction toward this intervention compared with a traditional visit was evaluated by using a Likert scale [14] ranging from 1 (the lowest satisfaction) to 5 (the highest satisfaction).

Finally, therapeutic adherence was measured through the Medication Adherence Rating Scale (MARS) [15,16] by dividing the categories of patients into adherence or not.

### 2.2. Statistical Analysis

Continuous variables are reported as mean values ± standard deviation (SD) or median values and interquartile intervals (IQR), whereas discrete variables are reported as frequencies and percentages. The Mann–Whitney U test for nonparametric variables and the chi-squared test for nominal variables were used. Multiple linear regression was used in order to evaluate the associations between these variables and the risk of GCA flares. Significance was fixed for a *p* level of 0.05 (95% confidence interval).

## 3. Results

A total of 74 remote monitoring visits were performed in 37 GCA patients included in the protocol. Patients were mostly women (77.8%) and had a mean age of 71.85 ± 9.25 years old. The mean disease duration was 5.8 ± 2.3 months. A total of 19 patients received oral glucocorticoids (GC) alone at the time of diagnosis with a mean daily dose of 52.7 ± 18.3 mg of prednisone (mean 0.98 mg/kg), while 18 patients were treated with a combination of oral steroids (at the time of diagnosis, prednisone mean dose was 51.7 ± 18.8 mg, mean 0.78 mg/kg) and subcutaneous injections of tocilizumab (TCZ) 162 mg weekly; the choice of therapy was made on the basis of the clinician’s opinion. Most patients had presenting cranial phenotype (28/37), characterized by headache, jaw claudication, and transient visual loss; others presented with symptomatic large vessel manifestation (9/37) characterized by constitutional symptoms such as fever, fatigue, and weight loss. In both groups of patients, there was reductions in ESR and CRP levels until normalization (Table 2). During the follow-up period, the ESR and CRP values remained, on average, within the normal range in both groups of patients analyzed with a minimal difference (no statistically significant, *p* > 0.05) concerning the ESR in favor of the patients treated with tocilizumab (Figure 1 and Figure 2). During the follow up, patients additionally treated with TCZ reduced their GC dose more than patients treated with GC alone (*p* = 0.03, Figure 3). Only one patient, who was treated with GC alone, had a cranial flare (meant as headache, scalp tenderness and temporal artery swelling) and needed to increase the dosage of GC from 10 to 25 mg of prednisone, who then showed rapid improvement. In general, the disease trend was consistent with the parameters recorded during telemedicine and then confirmed at the face-to-face visit. In particular, there was a progressive, statistically significant reduction in the inflammation indices of ESR and CRP (ESR Δ-61.06, *p*: 0.000; CRP Δ-46.39, *p* 0.000, respectively) from the time of diagnosis and then a stabilization of the clinical and laboratory picture, with no significant variations of the main parameters analyzed (ESR, CRP, PGA, and EGA; *p* > 0.05) during follow up (Table 2). Multivariate analysis showed no predictive factors predisposing to flare (ESR and CRP at the time of diagnosis, phenotype presentation, concomitant symptoms, and therapy; *p*-value > 0.05). Furthermore, all patients considered this type of monitoring very satisfactory according to a Likert scale, recording a mean score of 4.40 ± 0.21 (range 1–5) and the level of adherence to therapy was very high according to MARS (Table 3). All patients had a high adherence rate, and no significant difference emerged between the patient groups in the follow up.

Demographic data concerning the interviewed cohort at time of diagnosis, pre-lockdown visit (baseline), and during the remote follow-up are reported in Table 2.

## 4. Discussion

The results of this study demonstrated that telemedicine can be effectively and safely applied to monitor patients with GCA experience low disease activity as a possible alternative, at least for a limited period of time, to traditional visits. Our cohort consisted of patients with an established disease and therefore had sufficient experience with care self-management, whereas those who were diagnosed with GCA in March 2020 were excluded and followed by means of face-to-face visits.

Tools for the remote monitoring of patients with chronic diseases are widely available and the restrictions imposed by the COVID-19 pandemic have increased their use in clinical practice. Many medical fields have been invested in this “imposed” innovation. Among others, this strategy has, in fact, been adopted for rehabilitation [17,18] and the management of cardiovascular diseases [19,20,21], diabetes mellitus [22,23], neurological diseases [24,25,26], inflammatory bowel diseases [27], malignancies [28,29], cutaneous disorders [30], infectious diseases [31,32], and rheumatic diseases [33,34,35,36,37,38]. However, to the best of our knowledge, this is the first study aiming to examine the effectiveness of telemedicine for monitoring GCA patients.

GCA affects old and fragile patients, who represent a population with an intrinsic increased risk of severe COVID-19 [39]. Moreover, the basic treatment with GC for GCA further exposes these patients to SARS-CoV-2 infection and its complications.

In our analysis, one patient treated with GC only had a flare of disease, while no patient treated with tocilizumab had a flare. Although not significant, this finding is in line with those in the literature, which has shown that the addition of TCZ to prednisone facilitates earlier GCA control [40]. Indeed, treatment with TCZ allowed steroid tapering to the lowest possible dose faster than in patients treated with GC alone (*p*-value: 0.001).

Adherence to therapy was very high, and this was probably partly due to the relatively short observation period and partly due to the perception of the severity of the disease and the risk of blindness. However, one must always keep in mind that any measure of self-reported compliance overestimates compliance by approximately 30% [15].

An important consideration is that, given the difficulty experienced by the patients in accessing analysis and imaging laboratories due to the lockdown, we reduced the monitoring exams to a minimum. This certainly led to a closer follow-up but made it possible to avoid exposing patients to an additional risk of contagion.

It is important to continue the development and integration of technology based on the patient and on the dialogue between patients and healthcare professionals in order to allow digital evolution in an optimal way. With the help of digital technology, we will be able to offer high quality, affordable, and patient-centered personalized healthcare. Privacy and data protection are also of paramount importance in this regard, and an ideal healthcare system should evolve to use sensors that record data that are owned by the patient and that are collected in a patient-based system. Ideally, a framework for a patient-based platform should be defined in which governments define high standards of interoperability and data security and in which all data should be translated into the same interoperable language. It is therefore important that patients, healthcare professionals, hospitals, and companies can all speak the same digital language while using hardware and software according to their preferences. The patient would have ownership of the data and the possible will to share these data with healthcare professionals for research purposes.

Keeping in mind that elderly people usually have less confidence with informatic tools [41,42], we intentionally chose to interview patients by means of periodic video/phone calls, which proved to adequately cover the items provided by the EULAR minimal data set.

The limits of this study were the low number of patients recruited and the limited period of observation (4 months). Studies on wider cohorts conducted for longer periods of time should be therefore performed in order to confirm our results.

## 5. Conclusions

Our pivotal study demonstrates that GCA patients with low disease activity can be effectively and safely monitored and followed up by means of periodic video/phone call interviews. Though currently being applied to face an emergency situation, telemedicine could represent a valid alternative to traditional visits in the future for those patients with established chronic diseases, allowing for reductions in time and costs related to medical visits.

## Figures and Tables

**Figure 1 jpm-13-00620-f001:**
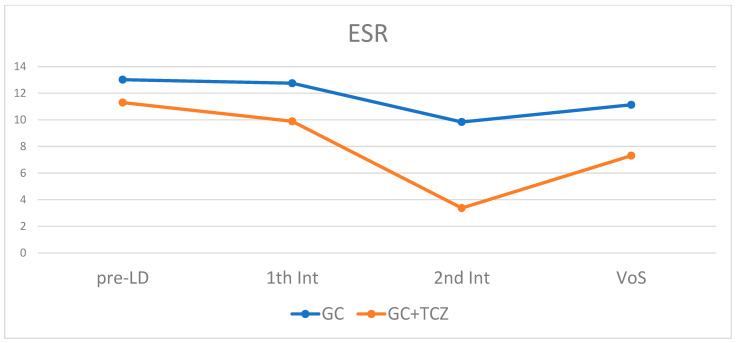
ESR trend during monitoring in the two different groups of patients treated with and without TCZ. ESR: erythrocyte sedimentation rate; GC: glucocorticoid; TCZ: tocilizumab; pre-LD: pre-lockdown visit; 1th Int: first interview; 2nd Int: second interview; VOS; visit on site.

**Figure 2 jpm-13-00620-f002:**
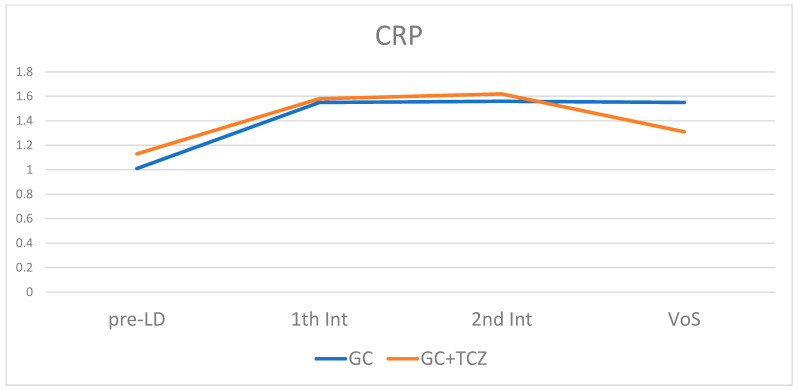
CPR trend during monitoring in the two different groups of patients treated with and without TCZ. CRP: C-reactive protein; GC: glucocorticoid; TCZ: tocilizumab; pre-LD: pre-lockdown visit; 1th Int: first interview; 2nd Int: second interview; VOS; visit on site.

**Figure 3 jpm-13-00620-f003:**
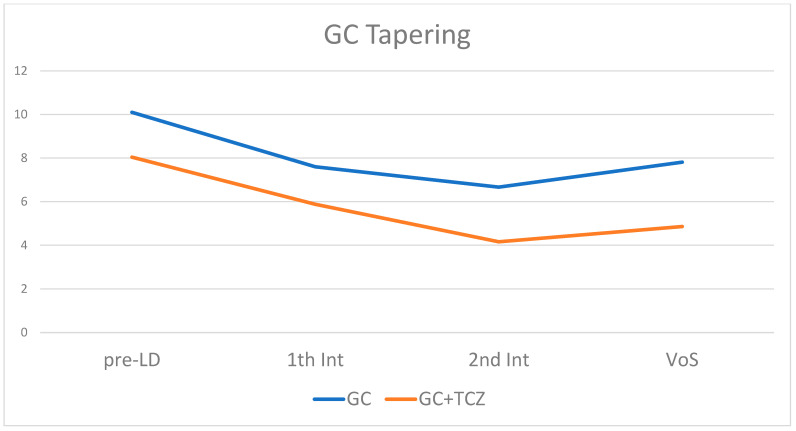
GC scaling in the two different groups of patients treated with and without TCZ. GC: glucocorticoid; TCZ: tocilizumab; pre-LD: pre-lockdown visit; 1th Int: first interview; VOS: visit on site.

**Table 1 jpm-13-00620-t001:** Domains and related information recorded during the interviews.

Domain	New Onset or Description of Concomitant Symptoms	Laboratory Tests	PROs	Therapy	Satisfaction	Compliance to Therapy
**Description**	- Fever - Blood pressure (if available) - Weight - Headache - Visual loss, diplopia or other ophthalmic manifestations - Jaw claudication - Scalp tenderness - Any contacts with the general practitioner and reasons why	- Blood count - Creatinine - Transaminases - ESR and CRP	- Patient global assessment - Evaluator global assessment	- Therapy for GCA - Therapy for concomitant diseases - Adverse events	0–5 Likert scale evaluating: - Global understanding of the interview; - Ease of listening to the speaker; - Quality of the assistance received through the interview; - convenience of the remote interview; - global satisfaction	Medication Adherence Rating Scale

GCA: giant cell arteritis; PROs: patient-reported outcomes.

**Table 2 jpm-13-00620-t002:** Disease course and remote monitoring.

Parameters	Time at Diagnosis	Pre-Lockdown	First Interview	Second Interview	Visit on Site
**Patients treated with only GC (19 pts)**					
**Leukocytes/mm^3^**	7330 (±930)	6820 (±980)	6700 (±910)	7640 (±1110)	8120 (±920)
**Platelets × 10^3^/mm^3^**	291 (±125)	324 (±111)	320 (±135)	330 (±102)	325 (±105)
**Creatinine mg/dL**	0.71 (±0.21)	0.75 (±0.19)	0.73 (±0.21)	0.71 (±0.24)	0.74 (±0.20)
**ALT U/L**	24.41 (±7.16)	22.10 (±9.20)	23.80 (±9.32)	26.22 (±7.88)	23.7 (±6.30)
**ESR** **(mean ± SD)**	74.08 (±23.95)	13.02 (±8.2)	12.75 (±6.42)	9.84 (±6.63)	11.13 (±7.45)
**CRP mg/L (mean ± SD)**	47.4 (±42.2)	1.01 (±1.1)	1.55 (±2.29)	1.5 (±2.24)	1.55 (±2.05)
**Hgb mg/dL (mean ± SD)**	11.6 (±2.6)	13.2 (±2.24)	12.75 (±3.19)	13.58 (±2.12)	13.74 (±0.55)
**PGA (1–10) median (IQR)**	8.5 (8–9)	3.5 (1–8)	3.5 (±1–7)	2 (1.5–8)	2 (1.25–4)
**EGA (1–10) median (IQR)**	7 (6.25–9)	2.5 (1–6)	3.5 (0.75–7)	2 (1–7)	1.5 (1.25–3)
**GC (PDN) mg (mean ± SD)**	52.72 (±18.3)	10.1 (±6.95)	7.6 (±4.51)	6.67 (±3.1)	7.81 (±4.32)
**Patients treated with GC and TCZ (18 pt)**					
**Leukocytes/mm^3^**	6330 (±820)	7120 (±730)	6600 (±1115)	7354 (±1003)	7120 (±720)
**Platelets × 10^3^/mm^3^**	361 (±135)	313 (±121)	330 (±145)	310 (±98)	315 (±109)
**Creatinine mg/dL**	0.74 (±0.22)	0.74 (±0.19)	0.81 (±0.31)	0.79 (±0.23)	0.76 (±0.30)
**ALT U/L**	25.11 (±6.22)	23.10 (±9.20)	24.82 (±8.18)	24.33 (±7.89)	25.51 (±5.20)
**ESR** **(mean ± SD)**	74.56 (±36.58)	11.3 (±3.85)	9.89 (±4.71)	3.37 (±4.4)	7.31 (±6.74)
**CRP mg/L (mean ± SD)**	47.8 (±41.7)	1.13 (±0.65)	1.58 (±0.85)	1.62 (±0.73)	1.31 (±1.95)
**Hgb mg/dL** **(mean ± SD)**	11.36 (±3.88)	12.98 (±4.91)	12.43 (±4.3)	13.25 (±5.15)	14.00 (±0.39)
**PGA (1–10) median (IQR)**	10 (6–10)	1 (1–7)	2 (1–4)	3 (1–4)	2 (1–3)
**EGA (1–10) median (IQR)**	9 (6–9)	1 (1–5)	1 (0.5–3)	3 (1–4)	2 (0–2)
**GC (PDN) mg (mean ± SD)**	51.76 (±18.85)	8.04 (±4.38)	5.88 (±3.92)	4.16 (±4.32)	4.46 (±3.56)

ALT: alanine transaminase; ESR: erythrocyte sedimentation rate; CRP: C-reactive protein; PGA: Patient Global Assessment; EGA: Evaluator Global Assessment; GC: glucocorticoid; PDN: prednisone; TCZ: tocilizumab; SD: standard deviation.

**Table 3 jpm-13-00620-t003:** Medication Adherence Rating Scale (MARS) assessed during the protocol.

Group of Patients	Pre-Lockdown	First Interview	Second Interview	Visit on Site
Overall, median (IQR)	8	7	8	7
	(6–9)	(6–9)	(6–9)	(6–9)
GC alone, median (IQR)	7	7	8	7
	(6–9)	(6–9)	(6–9)	(6–9)
GC+TCZ, median (IQR)	8	8	8	7.5
	(6–9)	(7–9)	(7–9)	(6–9)

GC: glucocorticoid; PDN: prednisone; TCZ: tocilizumab; IQR: interquartile range. MARS has a range from 1 to 10; ≥6: the patient is classified as adherent; <6: the patient is classified as not adherent

## Data Availability

Research data are available after reasonable request and ethics committee approval.

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
