# Peer review of "Management of Patients Affected by Giant Cell Arteritis during the COVID-19 Pandemic: Telemedicine Protocol TELEMACOV"

_jpm, 2023, doi:10.3390/jpm13040620_

Round 1

Reviewer 1 Report

Abstract:

“……were monitored monthly by means of video/phone calls from 9 March to 9 June 29

2020.”

Comment: In the patients and methods, it is stated that patients had remote interviews every 6-7 weeks. Please unify these statements in the abstract and in the patients and methods.

Results:

“A total of 148 remote monitoring visits were performed in 37 GCA patients included in the protocol.”

Comment: As far as I understand, that baseline and pre-lockdown visits and the last on site visits were face-to face, and only two visits were remote. Therefore the total number of remote visits cannot be 148.

“Most patients had presenting cranial phenotype (28/37), others with symptomatic large vessel manifestation (9/37).

Comment: Please show these details.

“Only one patient, who was treated with GC alone, had an ocular flare”

Comment: It is not clear how ocular flare could be diagnosed remotely without ophthalmological examination.

The term “baseline assessment” is misleading. Kindly use the term “time at diagnosis” instead in the main document and in the abstract. The baseline assessment in your study should be the pre-lockdown visit.

Please add the PGA and the EGA to table 1.

Laboratory tests mentioned in table 1 (serum creatinine, transaminases, total leucocytic count and platelet counts) are missing from table 2. Please add these to table 2.

“All patients had a disease course consistent with previous visits (telemedicine) at the follow up face to face visit.”

Comment: Please clarify your statement and support you statement with appropriate statistical testing.

“Multivariate analysis showed no predictive factors predisposing to flare (p-value>0.05).”

Comment: Please mention more details regarding the variables that were tested.

Discussion:

“The results of this study demonstrated that telemedicine can be effectively and safely applied to monitor patients with GCA as a possible alternative, at least for a limited period of time, to traditional visits”

Comment: Please note that your statement cannot be generalized. All patients had controlled disease and were using relatively low doses of corticosteroids, thus reducing the possibility of steroid-related adverse events. Telemedecine can be used safely only in patients with low disease activity as in this study.

Author Response

Dear Reviewer 1,

Thank you very much for the many observations and suggestions you have provided and which will surely contribute to making the paper better.

Below are our answers.

Abstract:

“……were monitored monthly by means of video/phone calls from 9 March to 9 June 29

2020.”

Comment: In the patients and methods, it is stated that patients had remote interviews every 6-7 weeks. Please unify these statements in the abstract and in the patients and methods.

Reply: We have modified these statements in the abstract and main text as suggested

Results:

“A total of 148 remote monitoring visits were performed in 37 GCA patients included in the protocol.”

Comment: As far as I understand, that baseline and pre-lockdown visits and the last on site visits were face-to face, and only two visits were remote. Therefore the total number of remote visits cannot be 148.

Reply: Correct observation. We have correctly recalculated the number of remote visits

“Most patients had presenting cranial phenotype (28/37), others with symptomatic large vessel manifestation (9/37).

Comment: Please show these details.

Reply: we have specified these subsets indicated the main manifestations

“Only one patient, who was treated with GC alone, had an ocular flare”

Comment: It is not clear how ocular flare could be diagnosed remotely without ophthalmological examination.

Reply: Thanks for this valuable comment. Indeed we realized that it is more appropriate to indicate as cranial flare and we have indicated the symptoms (especially in the absence of an ophthalmological evaluation)

The term “baseline assessment” is misleading. Kindly use the term “time at diagnosis” instead in the main document and in the abstract. The baseline assessment in your study should be the pre-lockdown visit.

Please add the PGA and the EGA to table 1.

Reply: thanks for this careful observation, we have corrected the term as suggested and add the PGA and EGA in table 1.

Laboratory tests mentioned in table 1 (serum creatinine, transaminases, total leucocytic count and platelet counts) are missing from table 2. Please add these to table 2.

Reply: we add these in table 2

“All patients had a disease course consistent with previous visits (telemedicine) at the follow up face to face visit.”

Comment: Please clarify your statement and support you statement with appropriate statistical testing.

“Multivariate analysis showed no predictive factors predisposing to flare (p-value>0.05).”

Comment: Please mention more details regarding the variables that were tested.

Reply: We rephrased the sentence more clearly by correlating it with statistical analysis and added the confounding variables analyzed in the multivariate analysis

Discussion:

“The results of this study demonstrated that telemedicine can be effectively and safely applied to monitor patients with GCA as a possible alternative, at least for a limited period of time, to traditional visits”

Comment: Please note that your statement cannot be generalized. All patients had controlled disease and were using relatively low doses of corticosteroids, thus reducing the possibility of steroid-related adverse events. Telemedecine can be used safely only in patients with low disease activity as in this study.

Reply: We have modified our statement as suggested

Thank you again for your time

All the best

Simone Parisi

Reviewer 2 Report

I read this manuscript with interest. These are my comments.

1. I think the paper explained the usefulness of video/phone call consultation, but I couldn't understand why blood was taken during this trial, even after reading Materials & Methods.

2. If blood was taken during the trial period and confirmed by the attending physician, I think it is possible that they influenced the treatment.

3. Figure 1 and Figure 2 are not explained in the text, so please cite and explain them in Results.

Author Response

Dear Reviewer 2,

Thank you very much for taking the time to do this careful review and for your suggestions

Below are our answers.

I read this manuscript with interest. These are my comments.

  1. I think the paper explained the usefulness of video/phone call consultation, but I couldn't understand why blood was taken during this trial, even after reading Materials & Methods.
  2. If blood was taken during the trial period and confirmed by the attending physician, I think it is possible that they influenced the treatment.

Reply 1 and 2: During the follow-up of the GCA, the patients, as per clinical practice and the recommendation of the guidelines, monitor the laboratory tests, in particular the inflammation indices. This is to evaluate whether the scaling of steroid therapy is adequate and contextual to the clinical control of the disease.  We have added an additional explanation in materials and methods on the reason for collecting data regarding blood sampling.

  1. Figure 1 and Figure 2 are not explained in the text, so please cite and explain them in Results.

Reply: we have explanined and cited figure 1 and 2 in the text

Thank you again for your time

All the best

Simone Parisi

Round 2

Reviewer 1 Report

Thank you for responding to all comments.

Reviewer 2 Report

All the questions raised by this reviewer were addressed appropriately.